# Malaria transmission in Africa: Its relationship with yellow fever and measles

**Oluyemi A. Okunlola**[1], **Oyetunde T. Oyeyemi**[2]*

**1** Department of Mathematics, University of Medical Sciences, Ondo, Ondo State, Nigeria, **2** Department of Biological Sciences, University of Medical Sciences, Ondo, Ondo State, Nigeria

* ooyeyemi@unimed.edu.ng

## Abstract

### Background

Malaria has been strongly linked to the transmission and pathophysiology of some viral diseases. Malaria and vaccine-preventable diseases often co-exist in endemic countries but the implication of their co-existence on their transmission dynamics and control is poorly understood. The study aims to evaluate the relationships between the incidence of malaria and cases of measles and yellow fever in Africa.

### Methods

The malaria incidence, death due to malaria, measles and yellow fever data were sourced from the WHO database. Poisson and zero-inflated time-trend regression were used to model the relationships between malaria and the two vaccine-preventable diseases. $P$-values <0.05 were considered statistically significant.

### Results

A significant negative relationship existed between malaria incidence and measles cases ($P$<0.05), however, malaria showed a positive relationship with yellow fever ($P$<0.05). The relationships between death due to malaria and measles/yellow fever cases followed similar trends but with a higher level of statistical significance ($P$<0.001).

### Conclusions

Malaria varied negatively with measles cases but positively with yellow fever. The relationships observed in this study could be important for the management of malaria and the studied vaccine-preventable diseases. Increase vaccination coverage and/or malaria treatment could modulate the direction of these relationships.

**Data Availability Statement:** The variables data examined in the study are available from the WHO database (www.who.int/data/gho).

**Funding:** The author(s) received no specific funding for this work.

## Introduction

Malaria is the most important parasitic disease in the tropical and sub-tropical regions of the world. In 2020, the World Health Organization (WHO) reported 241 million cases and 627

**Competing interests:** The authors have declared that no competing interests exist.

thousand deaths from malaria [1]; this is an increase from an estimated 227 million cases and 558 thousand deaths recorded in 2019 [2]. Africa is the most affected region of the world; accounting for more than 95% malaria cases and associated burdens, and deaths [2]. Currently, the public health intervention measures recommended by the WHO to control and prevent the transmission of malaria include; prompt detection of infection, treatment of clinical and severe malaria, vector control, and chemoprevention in high-risk groups [3].

Yellow fever like malaria, is a vector-borne disease caused by the yellow fever virus of the genus *Flavivirus* [3]. With estimated 51 000 to 380 000 severe cases of yellow fever, and 19 000 to 180 000 deaths per year in Africa [4], the continent is the epicenter of the disease recording 90% of the total global cases of the disease [5]. Approximately half of the infected populations are asymptomatic and a further third suffer from a mild illness. Only about 12% of yellow fever infections present a severe disease in the form of jaundice and/or haemorrhage with 30–60% death rate [6]. Measles, like malaria and yellow fever, is another disease with serious public health implications for children. Although the global transmission of measles has been greatly reduced by vaccination about 60 years ago, it continues to be one of the major causes of mortality among children in sub-Saharan Africa [7].

Previous studies have related malaria with some viral infections, especially HIV and hepatitis [8–10], and these studies suggested the possible impact of malaria on the dynamics of transmission and morbidities associated with viral infections. Despite the dreadfulness of malaria and viral agents in the current study, studies on concomitant presentation of these diseases are very rare. Although few epidemiological studies are available for malaria and yellow fever [11,12], none is available for malaria and measles. Other studies involving the triads included the effect of chloroquine prophylaxis on yellow fever virus replication and vaccine antibody response [13,14], and impact of placental malaria on infant immunity to measles [15]. An experimental data indicated that chloroquine may inhibit yellow fever virus in vitro [13], but the only available clinical evidence showed that chloroquine preventive treatment against malaria had no effect on antibody response to yellow fever vaccination [14]. In a similar vein, malaria during pregnancy was reported to impair the transplacental transfer of measles antibody, thus, suggesting that malaria control during pregnancy could have substantial benefits in the mother and the baby including reduction in the risk of measles in early infancy [15].

Considering the similar geographical distributions between malaria and vaccine preventable diseases (yellow fever virus and measles) and the possible associations that could exist between them, analysis of relationships between them using robust secondary data could provide more information on the trend of these relationships. The relationships could inform policy on what approach could be best adopted for the various preventive or interventional programmes related to these diseases.

The aim of the current study is therefore to determine the magnitude and direction of the trend of malaria incidence, death due to malaria, measles cases and yellow fever cases in selected African countries. Also to investigate the effect of measles and yellow fever cases on malaria incidence and death due to malaria, respectively.

## Material and methods

### Data and scope

Available data on malaria incidence, death due to malaria, measles and yellow fever was downloaded from public domain (https://www.who.int/data/gho) for all African countries. Countries with a well filled data were arranged in a separate Microsoft excel sheet for each variable which produced 47 countries for malaria incidence, 46 countries for death due to malaria, 54 countries for measles cases and 22 countries for yellow fever cases. Each of the Microsoft excel

sheets was imported to STATA software package for trend analysis. Prior to estimation of regression model, the data was merged into two STATA data file. The first file contains malaria incidence, measles and yellow fever while the second contain death due to malaria, measles cases and yellow fever cases. We ensured that only information with uniform country and time were retained in the merged files. The malaria incidence and death due to malaria files used regression had 49 countries (2000–2018) and 46 countries (2000–2017), respectively.

## Regression analyses

Count data are known to be discrete variables and not continuous, thus, the traditional regression model that assumes that outcome or response variable comes from a normal distribution is not tenable bearing in mind that discrete variable is highly skewed. This study used a discrete probability distribution that treats the response variable of count observation as a Poisson process with probability mass function given as

$$f(y) = \frac{e^{-\mu}\mu^{y}}{y!}, 0 \leq y \leq \infty \tag{1}$$

This bound indicates that the distribution takes on nonnegative random numbers. The quantity $\lambda$ is the mean of the Poisson distribution. This distribution has a unique property of equi-dispersion that the mean, $\lambda = E(y)$ is equal to variance, $V(y)$. However, the assumption of equi-dispersion is highly restrictive and hardly satisfied in real-life application as shown here that the mean each variable considered is not equal to the variance. Two major problems that are associated with Poisson regression are over-dispersion and under-dispersion which connote situations where $V(y) > E(y)$ and $V(y) < E(y)$, respectively. Sellers and Shmueli [16] provided a more general count distribution that captured a wide range of dispersion which they titled Conway–Maxwell-Poisson (COM-Poisson) and Huang [17] provided a COM-Poisson Double Generalized Linear Model that addressed the problem of both under and overdispersion simultaneous.

Poisson regression is achieved by expressing $\lambda$ in Eq (1) as a function of some explanatory variables through a log link function. This can be mathematically expressed as:

$$\lambda_i = X'\beta \text{ or } \quad \mu_i = \exp(X'\beta) \tag{2}$$

Given the density function of Eq (1) type and set of independent variable $X_i$, the maximum likelihood estimates of regression parameter $\beta$ is defined as;

$$log_e L(\beta) = \sum \{Y_i X'\beta - \exp(X'\beta) - \ln(Y!)\} \tag{3}$$

Count data most often have excess zero cases and if not adjusted for can lead to inaccurate regression estimate and misleading conclusion. Lambert [18] developed a zero-inflated Poisson (ZIP) regression that corrects for the occurrence of excess zero cases in count data. The ZIP has become a popular choice in empirical problems relating to count data. The ZIP model is often appealing, first, since it divides the dependent variable into two subpopulations where the first one takes on the value of zero with probability $\pi_i$ while the second one is Poisson distributed with probability $1 - \pi_i$ [19]. Due to this setup, the ZIP model may be used when the data contain an excess amount of zeros. Second, this model is popular because it accounts for overdispersion (meaning that the variance of the dependent variable exceeds the mean value), which leads to an underestimation of the variance of the estimated coefficients when applying the standard Poisson regression model [19]. The ZIP divides the count data into two non-

overlapping subpopulations using the equation:

$$y_i = \begin{cases} 0 & \text{with porbability } \pi_i \\ P_0(\mu_i) & \text{with porbability } 1 - \pi_i \end{cases} \text{ and } \pi_i = \frac{\exp(q_i'\gamma)}{1 + \exp(q_i'\gamma)} \tag{4}$$

In this expression $q_i$ is the i[th] row of the Q, which is the data matrix for the logit model and $\gamma$ corresponds to a $(p + 1) \times 1$ vector of coefficients. Also, $\mu_i = \exp(x_i'\beta)$, where $x_i$ is the i[th] row of X, which is the data matrix for the ZIP model while $\beta$ corresponds to a $(p + 1) \times 1$ vector of coefficients. The most common method of estimating $\beta$ is to apply the ML method. By defining $1(yi = 0)$ as an indicator function that takes on the value of 1 if $y_i = 0$, the following joint likelihood should then be maximized:

$$L(\beta, \gamma) = \sum_{i=1}^{N} (l(y_i = 0)(\log(\exp(q_i'\gamma)) + \exp(-\exp(-\exp(X_i'\beta))))$$
$$+ \sum_{i=1}^{N} (1 - l(y_i = 0)(y_i', x_i'\beta - \exp(x_i'\beta - \exp(x_i'\beta))] - \sum_{i=1}^{N} \log(1 + \exp(q_i'\gamma)) \tag{5}$$

This complex likelihood function is estimated through the modification of the simplex method of Nelder and Mead [20] proposed by Kibria et al. [19] in which the start-up values come from the iterative weighted least-squares algorithm of the individual Poisson and logit estimation. The modification to this method applied the estimator specified in (6).

$$\hat{\beta}_{RR} = (X`\hat{W}X + kl)^{-1}X`\hat{W}X\hat{\beta}_{ML} \tag{6}$$

$\hat{\beta}_{ML}$ is the estimate obtained using the simplex method. Furthermore, the shrinkage parameter $k$ may take on values between zero and infinity, and when $k$ equals zero, we have $\hat{\beta}_{RR} = \hat{\beta}_{ML}$. When $k$ is greater than zero, we have $||\hat{\beta}_{RR}|| < ||\hat{\beta}_{ML}||$. Since $\hat{\beta}_{ML}$ is, on average, too long in the presence of multicollinearity, $\hat{\beta}_{RR}$ is expected to perform better than $\hat{\beta}_{ML}$.

## Statistical analysis

The statistical analysis was carried out using five statistical software packages. SPSS version 23 was used for data management, cleaning and descriptive analysis. STATA version 13 and R version 4.0.3 were used to estimate the Poisson and zero-inflated Poisson (ZIP) regression model. Because all the studied variables are cases and rates which are regarded as count data, the Panel Poisson regression model was adopted to understudy these relationships. Where there are observed excess zero in the cases of event considered in the study, the zero-inflated Poisson (ZIP) regression that is robust to the occurrence of zero was adopted to achieve our specified objectives. GeoDA 1.9 was used to prepare the shapefile and merging of the estimated annual average change (EAAC) while QGIS version 3.14 was used for geospatial mapping of the EAAC of each variable. $P$-values $<0.05$ were considered statistically significant.

## Results

Time trend Poisson and zero-inflated Poisson model was fitted for malaria incidences (47 countries), death due to malaria (46 countries), measles cases (54 countries) and yellow fever cases (22 countries). Poisson regression was used for 46(97.87%), 46(100.0%) and 47(87.04%) for malaria incidence, death due to malaria and measles cases while zero-inflated Poisson was used in 1(2.13%), 7(12.06%) and 22(100.0%) countries where there were clear occurrences of excess zero in the malaria incidence, measles cases and yellow fever cases respectively.

Malaria incidence reduced significantly in 38(80.9%) countries, increased significantly in 5 (10.6%) and increased insignificantly in 4(8.51%) of out the 47 countries observed in the study over time. Death due to malaria decreased significantly in 20(43.5%) countries but increased

significantly in 24(52.2%) of the 46 countries included for the study over time. Also, of the total 54 countries studied for measles cases, 49(90.7%) showed significant decrease trend, 4 (7.4%) significant increase trend and 1(1.85%) insignificant time trend coefficient. Yellow fever cases decreased significantly in 13(59.1%), increased significantly in 3(13.6%) countries while the remaining countries 6(27.27%) showed an insignificant trend (Table 1).

The spatial distribution of malaria incidence in African countries is presented in Fig 1. Malaria incidence average annual trend decreased significantly in several African countries except in Algeria, Cape Verde, Mali and Eswatini with time trend regressions -0.088, -0.016, -0.002 and -0.037, respectively, which showed a non-significant trend (*P*>0.05). A significant increase in malaria incidence was recorded in Djibouti (0.144), Eritrea (0.046), Madagascar (0.024), Rwanda (0.078), and Niger (0.008) (*P*<0.05) (Table 1).

The spatial distribution of malaria-associated death in African countries is presented in Fig 1. Algeria with trend regression (0.257) recorded the highest increase in death due to malaria while Egypt (-0.514), on the other hand, recorded the highest decrease in death due to malaria (*P*<0.05). Several countries in East Africa recorded a significant increase in malaria-associated death while many countries in Central Africa however, showed a significant decrease in malaria-associated death (*P*<0.05). All the countries in the South African region recorded a significant increase in death due to malaria (*P*<0.05) (Table 1).

The spatial distribution of measles in African countries is presented in Fig 2. Majority of the African countries recorded a significant decrease in measles cases except in Mauritius (0.042), Somalia (0.063), South Sudan (0.058), and Democratic Republic of Congo (0.062) (*P*<0.05). Seychelles (-0.006) showed a non-significant decrease in measles cases annual trend (*P*>0.05) (Table 1). The spatial distribution of yellow fever cases is presented in Fig 2. Chad with trend regression 0.268 recorded the highest significant increase in yellow fever cases while Congo (-0.236) showed the highest significant decrease in yellow fever cases in Africa (*P*<0.05). The Central Africa Republic (-0.017), Burkina Faso (0.003), Côte d'Ivoire (0.004), Guinea-Bissau (0.03), Mauritania (-0.047), and Togo (0.039) recorded non-significant yellow fever cases trends (Table 1).

The Panel Poisson Regression presented in Table 2 showed the effect of Measles and Yellow Fever in each of African Region and Africa as a whole. Malaria incidence increased significantly with measles cases in East Africa while significant negative relationship occurred between the two in West and South African countries (*P*<0.001). Overall, a significant negative relationship existed between malaria incidence and measles cases (*P*<0.05). While a positive relationship existed between malaria incidence and yellow fever cases in West Africa (*P*<0.01), a negative relationship occurred between the two in South Africa (*P*<0.001) (Table 2). Overall, malaria incidence was significantly positively related to yellow fever in Africa (*P*<0.05).

Death due to malaria was significantly negatively related to measles cases in Africa (*P*<0.001). A negative relationship existed between the two West and South African countries (*P*<0.001). Overall, a significant positive relationship existed between death due to malaria and yellow fever in Africa (*P*<0.001) (Table 2).

## Discussion

Malaria is still being actively transmitted across African countries, however, observations from our study showed that the various interventional programmes could be yielding expected results as there was a significant decrease in malaria incidence in about 81% of the observed African countries. Increase insecticide-treated nets (ITNs) coverage across African countries over the years may be responsible for this malaria decline [21,22]. The use of ITNs may result

Table 1. Poisson and zero inflated time trend regression.

| Region | Country | Malaria Incidence | | | Death due to Malaria | | | Measles Cases | | | Yellow fever Cases | | | |
|---|---|---|---|---|---|---|---|---|---|---|---|---|---|---|
| | | Method | Estimate | Significance | Method | Estimate | Significance | Method | Estimate | Significance | Method | Estimate | YFTD | Significance |
| North | Algeria | P | -0.088 | 0.901 | P | 0.257 | 0.023 | P | -0.082 | < 0.001 | | | | |
| North | Egypt | | | | P | -0.514 | 0.025 | P | -0.045 | < 0.001 | | | | |
| North | Libya | | | | | | | P | -0.041 | < 0.001 | | | | |
| North | Morocco | ZIP | -0.405 | 0.976 | P | 0.13 | 0.383 | P | -0.17 | < 0.001 | | | | |
| North | Sudan | P | -0.046 | < 0.001 | P | -0.144 | < 0.001 | P | -0.111 | < 0.001 | | | | |
| North | Tunisia | | | | | | | P | -0.064 | < 0.001 | | | | |
| East | Burundi | P | -0.05 | < 0.001 | P | -0.171 | 0.021 | P | -0.135 | < 0.001 | | | | |
| East | Comoros | P | -0.084 | < 0.001 | | | | ZIP | -0.087 | < 0.001 | | | | |
| East | Djibouti | P | 0.144 | < 0.001 | P | -0.055 | 0.087 | P | -0.117 | < 0.001 | | | | |
| East | Eritrea | P | 0.046 | < 0.001 | P | -0.018 | 0.003 | P | -0.061 | < 0.001 | | | | |
| East | Ethiopia | P | -0.074 | < 0.001 | P | 0.093 | < 0.001 | P | -0.039 | < 0.001 | | | | |
| East | Kenya | P | -0.08 | < 0.001 | P | 0.165 | < 0.001 | P | -0.175 | < 0.001 | ZIP | -0.119 | 1 | 0.007 |
| East | Madagascar | P | 0.024 | < 0.001 | P | 0.019 | < 0.001 | P | -0.042 | < 0.001 | | | | |
| East | Malawi | P | -0.036 | < 0.001 | | | | P | -0.126 | < 0.001 | | | | |
| East | Mauritius | | | < 0.001 | | | | ZIP | 0.042 | < 0.001 | | | | |
| East | Mozambique | P | -0.025 | < 0.001 | P | 0.125 | < 0.001 | P | -0.066 | < 0.001 | | | | |
| East | Rwanda | P | 0.078 | < 0.001 | P | 0.168 | < 0.001 | P | -0.168 | < 0.001 | | | | |
| East | Seychelles | | | | | | | ZIP | -0.006 | 0.084 | | | | |
| East | Somalia | P | -0.104 | < 0.001 | P | 0.078 | < 0.001 | P | 0.063 | < 0.001 | | | | |
| East | South Sudan | P | -0.029 | < 0.001 | P | -0.255 | < 0.001 | P | 0.058 | < 0.001 | | | | |
| East | Uganda | P | -0.038 | < 0.001 | P | -0.011 | < 0.001 | P | -0.04 | < 0.001 | ZIP | -0.071 | 1 | < 0.001 |
| East | United Republic of Tanzania | P | -0.07 | < 0.001 | P | 0.013 | < 0.001 | P | -0.139 | < 0.001 | | | | |
| East | Zambia | P | -0.049 | < 0.001 | P | 0.101 | < 0.001 | P | -0.128 | < 0.001 | | | | |
| East | Zimbabwe | P | -0.033 | < 0.001 | P | 0.145 | < 0.001 | ZIP | -0.077 | < 0.001 | | | | |
| Central | Angola | P | -0.035 | < 0.001 | P | 0.035 | < 0.001 | P | -0.084 | < 0.001 | ZIP | 0.11 | 2 | < 0.001 |
| Central | Cameroon | P | -0.034 | < 0.001 | P | -0.026 | < 0.001 | P | -0.107 | < 0.001 | ZIP | -0.032 | 1 | < 0.001 |
| Central | Central African Republic | P | -0.016 | < 0.001 | P | -0.165 | < 0.001 | P | -0.065 | < 0.001 | ZIP | -0.017 | 1 | 0.605 |
| Central | Congo | P | -0.033 | < 0.001 | P | -0.078 | < 0.001 | P | -0.116 | < 0.001 | ZIP | -0.236 | 1 | < 0.001 |
| Central | Democratic Republic of the Congo | P | -0.028 | < 0.001 | P | -0.075 | < 0.001 | P | 0.062 | < 0.001 | ZIP | -0.136 | 1 | < 0.001 |
| Central | Equatorial Guinea | P | -0.02 | < 0.001 | P | -0.161 | < 0.001 | P | -0.101 | < 0.001 | | | | |
| Central | Gabon | P | -0.007 | 0.017 | P | 0.192 | < 0.001 | P | -0.085 | < 0.001 | ZIP | -0.042 | 1 | < 0.001 |
| Central | Sao Tome and Principe | P | -0.213 | < 0.001 | P | 0.314 | < 0.001 | ZIP | -0.088 | < 0.001 | | | | |
| West | Benin | | | | | | | P | -0.099 | < 0.001 | ZIP | -0.061 | 1 | 0.002 |
| West | Burkina Faso | P | -0.02 | < 0.001 | P | -0.006 | < 0.001 | P | -0.056 | < 0.001 | ZIP | 0.003 | 2 | 0.266 |
| West | Cape Verde | P | -0.016 | 0.785 | P | 0.088 | < 0.001 | P | -0.077 | < 0.001 | | | | |
| West | Chad | P | -0.018 | < 0.001 | P | -0.109 | < 0.001 | P | -0.033 | < 0.001 | ZIP | 0.268 | 2 | < 0.001 |
| West | Côte d'Ivoire | P | -0.03 | < 0.001 | P | 0.045 | < 0.001 | P | -0.075 | < 0.001 | ZIP | 0.004 | 2 | 0.408 |
| West | Gambia | P | -0.068 | < 0.001 | P | 0.036 | < 0.001 | ZIP | -0.085 | < 0.001 | | | | |

(Continued)

**Table 1.** (Continued)

| Region | Country | Malaria Incidence | | | Death due to Malaria | | | Measles Cases | | | Yellow fever Cases | | | |
|---|---|---|---|---|---|---|---|---|---|---|---|---|---|---|
| | | Method | Estimate | Significance | Method | Estimate | Significance | Method | Estimate | Significance | Method | Estimate | YFTD | Significance |
| West | Ghana | P | -0.027 | < 0.001 | P | 0.034 | < 0.001 | P | -0.097 | < 0.001 | ZIP | -0.033 | 1 | < 0.001 |
| West | Guinea | P | -0.013 | < 0.001 | P | -0.042 | < 0.001 | P | -0.057 | < 0.001 | ZIP | -0.069 | 1 | < 0.001 |
| West | Guinea-Bissau | P | -0.096 | < 0.001 | P | 0.066 | < 0.001 | P | -0.075 | < 0.001 | ZIP | 0.03 | 2 | 0.301 |
| West | Liberia | P | -0.018 | < 0.001 | P | -0.079 | < 0.001 | P | -0.065 | < 0.001 | ZIP | -0.22 | 1 | < 0.001 |
| West | Mali | P | -0.002 | 0.312 | P | -0.036 | < 0.001 | P | -0.089 | < 0.001 | ZIP | -0.072 | 1 | < 0.001 |
| West | Mauritania | P | -0.075 | < 0.001 | P | -0.035 | < 0.001 | P | -0.106 | < 0.001 | ZIP | -0.047 | 1 | 0.056 |
| West | Niger | P | 0.008 | < 0.001 | P | -0.023 | < 0.001 | P | -0.048 | < 0.001 | | | | |
| West | Nigeria | P | -0.025 | < 0.001 | P | -0.02 | < 0.001 | P | -0.05 | < 0.001 | ZIP | -0.052 | 1 | < 0.001 |
| West | Senegal | P | -0.106 | < 0.001 | P | 0.086 | < 0.001 | P | -0.099 | < 0.001 | ZIP | -0.026 | 1 | < 0.001 |
| West | Sierra Leone | P | -0.01 | < 0.001 | P | -0.108 | < 0.001 | P | -0.056 | < 0.001 | ZIP | 0.151 | 2 | < 0.001 |
| West | Togo | P | -0.026 | < 0.001 | P | 0.013 | < 0.001 | P | -0.149 | < 0.001 | ZIP | 0.039 | 2 | 0.080 |
| South | Botswana | P | -0.188 | < 0.001 | P | 0.077 | < 0.001 | P | -0.099 | < 0.001 | | | | |
| South | Eswatini | P | -0.037 | 0.265 | P | 0.165 | < 0.001 | ZIP | -0.122 | < 0.001 | | | | |
| South | Lesotho | | | | | | | P | -0.135 | < 0.001 | | | | |
| South | Namibia | P | -0.095 | < 0.001 | P | 0.29 | < 0.001 | P | -0.136 | < 0.001 | | | | |
| South | South Africa | P | -0.063 | 0.030 | P | 0.019 | < 0.001 | P | -0.096 | < 0.001 | | | | |

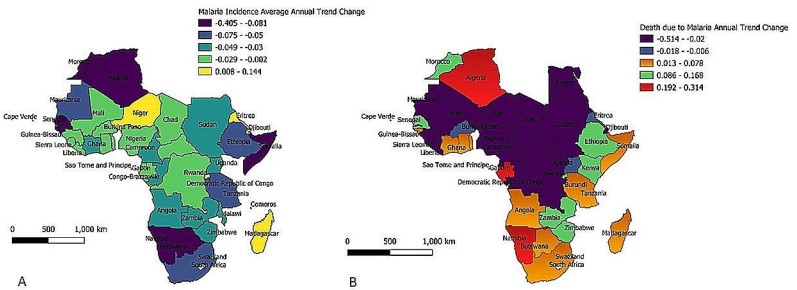

**Fig 1. Spatial Distribution of Malaria (A) and Death due to Malaria (B) in Africa.**

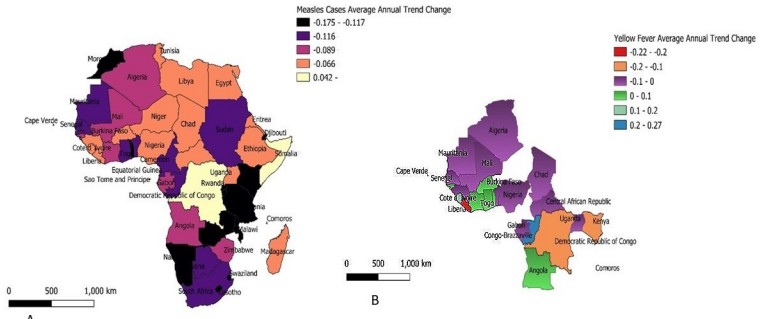

**Fig 2. Spatial Distribution of Measles (A) and Yellow Fever Cases (B) in Africa.**

**Table 2. Malaria incidence and death due to random effect panel model.**

| Variables | | North | East | Central | West | South | Overall |
|---|---|---|---|---|---|---|---|
| Malaria incidence | Measles | 0.0000074 | 0.0000053*** | 0.00000015 | -0.00000084*** | -0.00057*** | -0.00000042** |
| | | (0.52) | (5.73) | (0.65) | (-4.24) | (-6.21) | (-2.86) |
| | Yellow Fever | 0 | 0.000346 | 0.0000428 | 0.000114** | -0.548*** | 0.0000786** |
| | | (.) | (1.27) | (1.22) | (3.27) | (-4.42) | (3.20) |
| | trend | -0.0356*** | -0.0224*** | -0.0308*** | -0.0237*** | -0.0762*** | -0.0262*** |
| | | (-6.44) | (-16.11) | (-33.47) | (-34.84) | (-8.11) | (-52.68) |
| | Constant | 4.691*** | 5.472*** | 5.960*** | 5.956*** | 3.107*** | 5.653*** |
| | | (4.15) | (22.07) | (45.80) | (25.24) | (4.74) | (30.10) |
| | Observations | 27 | 123 | 152 | 264 | 43 | 609 |
| Death due to malaria | Measles | -0.000016 | -0.000014*** | -0.0000031*** | -0.0000014*** | -0.00038*** | -0.0000043*** |
| | | (-0.92) | (-59.09) | (-77.37) | (-16.74) | (-20.32) | (-125.04) |
| | Yellow fever | 0 | 0.00137*** | 0.0000890*** | -0.0000127 | -0.356*** | 0.000156*** |
| | | (.) | (15.31) | (9.79) | (-1.10) | (-12.81) | (20.57) |
| | Trend | -0.0790*** | -0.129*** | 0.00648*** | 0.00381*** | -0.172*** | -0.0472*** |
| | | (-3.86) | (-377.34) | (20.67) | (11.20) | (-59.53) | (-270.38) |
| | Constant | 5.748*** | 9.715*** | 8.456*** | 7.560*** | 6.553*** | 8.567*** |
| | | (4.48) | (22.44) | (16.29) | (24.47) | (10.05) | (33.46) |
| | Observations | 15 | 88 | 118 | 224 | 39 | 484 |

$t$ statistics in parentheses.

* $p < 0.05$,

** $p < 0.01$,

*** $p < 0.001$.

in a decline in the population of indoor *Anopheles* mosquitoes by triggering behavioural changes in biting and resting of mosquitoes including a shift from an indoor to an outdoor environment [23]. Besides the increase in ITNs coverage, changes in socio-ecological conditions in many malaria-endemic areas of Africa including changes in temperature, deforestation [24], increase use of agricultural pesticides with mosquitocidal properties [25], improved house constructions [26] could result in a decreased density of mosquitoes.

It is however important to stress that the recent WHO published report showed increase in malaria cases and associated deaths after 2019 [27]. The reasons for this have been accrued to plateaued malaria interventional funding, increase in drug and insecticide resistance, and climate change which threatens to push malaria transmission to new regions [28]. COVID-19 has further disrupted progress and implementation of various malaria intervention programs [28].

The significant decrease in malaria transmission could have culminated in the 43.5% significant reduction in death associated with malaria in Africa. However, the 52.2% significant increase in malaria-associated deaths in Africa could have resulted from poor access to and low-quality health services, and increased resistance of *Plasmodium falciparum* to first-line drugs [29,30]. While the incidence of malaria may correlate with death associated with the disease, the malaria-associated deaths could be difficult to assess in the actual sense [31,32].

The higher significant decrease in the incidence of measles in many African countries compared with that of yellow fever could portray a probable relaxed public health alert system towards prevention of yellow fever. Importantly, the immunization coverage of yellow fever appeared to be lower than that of measles in many African countries. For example in Nigeria, the mean official yellow fever vaccine (YFV) coverage was 56.3% while that of the first dose of measles-containing vaccine (MCV1) was 65.0%, for the Democratic Republic of Congo, YFV and MCV1 were 80.67 and 87.67% respectively [33,34]. In Togo however, the mean immunization coverage for YFV (82.83%) and MCV1 (82.75%) was similar [35].

While our models showed a generalized negative relationship between the incidence of malaria and measles cases, the factors responsible for this relationship cannot be readily explained at the moment. The usual administration of preventive chemotherapy such as chloroquine in malaria highly endemic areas of Africa may inhibit the replication of measles virus in concomitant infection, thus, the observed negative relationship. One study in Ghanaian pregnant women population has also associated maternal malaria preventive treatment with increasing infant immunity against measles [14]. A closer look at the differences in the direction of the relationship at the regional levels however suggested that in Eastern Africa, malaria transmission could impact measles transmission. While the reason for these conflicting observations in Eastern Africa cannot be readily explained at the moment, well designed clinical studies in all the regions could resolve these conflicting observations. Other unknown factors other than preventive chemotherapy could play some roles.

The overall positive relationship between malaria and yellow fever could be as a result of co-existence of mosquito vectors of the two diseases in similar ecological zones. In addition to *Aedes* mosquitoes, *Anopheles* mosquitoes have been identified in similar habitats in Africa [36,37]. Although our study showed decreased in the incidence of malaria over the years in many countries in West Africa, the African region is one of the most affected regions of the world contributing more than 30% of the world's malaria-associated deaths [38]. This could be difficult to explain, but complex immune modulation which could influence the susceptibility and immunopathology of yellow fever in malaria-endemic areas may be responsible for the positive relationship in the incidence of the two diseases in the region. The reverse relationship trend between the two diseases in countries in the Southern Africa region may be due to lower malaria transmission in many countries in the region. Overall similar relationship trends

observed between malaria incidence and yellow fever/measles were also observed between death due to malaria and the two vaccine-preventable diseases in Africa. These can be adduced to the aforementioned reasons highlighted for malaria incidence and yellow fever/measles.

## Conclusions

Our study showed that significant relationships exist between malaria and yellow fever/measles. The direction of the relationships differed, with yellow fever showing positive and measles negative relationships with malaria incidence and malaria-related deaths, respectively. The relationships observed in this study could be important for the management of malaria and the studied vaccine-preventable diseases. Increase vaccination coverage or malaria treatment could shape the direction of these relationships. Case-control studies are recommended to critically understudy the relationships between malaria, measles and yellow fever.

## Supporting information

**S1 Data.**
(XLSX)

## Author Contributions

**Conceptualization:** Oyetunde T. Oyeyemi.

**Data curation:** Oluyemi A. Okunlola, Oyetunde T. Oyeyemi.

**Formal analysis:** Oluyemi A. Okunlola.

**Investigation:** Oluyemi A. Okunlola, Oyetunde T. Oyeyemi.

**Methodology:** Oluyemi A. Okunlola, Oyetunde T. Oyeyemi.

**Writing – original draft:** Oluyemi A. Okunlola, Oyetunde T. Oyeyemi.

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
