## [Decision Letter · Decision Letter 0]

5 Jan 2022

PONE-D-21-27941Malaria transmission in Africa: its relationship with yellow fever and measlesPLOS ONE

Dear Dr. Oyeyemi,

Thank you for submitting your manuscript to PLOS ONE. After careful consideration, we feel that your manuscript will likely be suitable for publication if the authors revise it to address specific points raised  by the reviewer. According to the reviewers, there are some specific areas where further improvements would be of substantial benefit to the readers.   For your guidance, a copy of the reviewers' comments was included below.  

We look forward to receiving your revised manuscript.

Kind regards,

Luzia Helena Carvalho, Ph.D.

Academic Editor

PLOS ONE

https://journals.plos.org/plosone/s/file?id=ba62/PLOSOne_formatting_sample_title_authors_affiliations.pdf”

4. We note that Figure(s) 1 and 2 in your submission contain [map/satellite] images which may be copyrighted. All PLOS content is published under the Creative Commons Attribution License (CC BY 4.0), which means that the manuscript, images, and Supporting Information files will be freely available online, and any third party is permitted to access, download, copy, distribute, and use these materials in any way, even commercially, with proper attribution. For these reasons, we cannot publish previously copyrighted maps or satellite images created using proprietary data, such as Google software (Google Maps, Street View, and Earth). For more information, see our copyright guidelines: http://journals.plos.org/plosone/s/licenses-and-copyright.

 a. You may seek permission from the original copyright holder of Figure(s) 1 and 2 to publish the content specifically under the CC BY 4.0 license. 

Reviewers' comments:

Reviewer's Responses to Questions

**Comments to the Author**

1. Is the manuscript technically sound, and do the data support the conclusions?

Reviewer #1: Yes

2. Has the statistical analysis been performed appropriately and rigorously? 

Reviewer #1: Yes

3. Have the authors made all data underlying the findings in their manuscript fully available?

Reviewer #1: Yes

4. Is the manuscript presented in an intelligible fashion and written in standard English?

Reviewer #1: Yes

5. Review Comments to the Author

Reviewer #1: General comments

This is an important subject in Africa, the continent with the highest burden of malaria, but of recently experiencing outbreaks due to viral diseases.

Line 69: Authors should change the word “Precious” to “Previous”

Line 207. The statement is not correct. Rephrase to read “Fever in each of African Region and Africa as a whole”

Line 236: The word ITN has been introduced without being defined anywhere in the manuscript

Lines 236-237: The statement must be supported by appropriate References

Line 276: The mention of Culex mosquito is irrelevant and confusing. The mosquito species neither transmit malaria no Yellow fever.

6. PLOS authors have the option to publish the peer review history of their article (what does this mean?). If published, this will include your full peer review and any attached files.

Reviewer #1: **Yes: **Leonard Mboera

---

## [Author Response · Author response to Decision Letter 0]

6 Jan 2022

Response to comments

We thank you for your efforts both in giving our manuscript quality review and expediting the process. As suggested and requested by the Editor and the Reviewer, we have provided responses to the concerns raised. Our responses are appended below:

Editor’s comments

1. We have now adapted the manuscript as suggested to the journal’s style

2. Online profile is now update to include the ORCID number of the corresponding

3. We have now removed the phrase “data not shown” and associated sentence from the manuscript since the data are not a core part of the research being presented in our study.

4. Figures 1 and 2 are original and we generated by the software QGIS (version 3.14). This was stated in the statistical analyses. So, they have no copyright issues.

5. Data availability statement has been revised and the website included. The file containing the minimal data is also included as supplementary file.

Reviewer #1: General comments

This is an important subject in Africa, the continent with the highest burden of malaria, but of recently experiencing outbreaks due to viral diseases.

Line 69: Authors should change the word “Precious” to “Previous”

Response – Thank you for the observation. It has now been corrected

Line 207. The statement is not correct. Rephrase to read “Fever in each of African Region and Africa as a whole”

Response – Thank you. This is now corrected as requested

Line 236: The word ITN has been introduced without being defined anywhere in the manuscript

Response – Thank you for the observation, it is now defined at first mention (please see line 224)

Lines 236-237: The statement must be supported by appropriate References

Response – Thank you, we have now included references

Line 276: The mention of Culex mosquito is irrelevant and confusing. The mosquito species neither transmit malaria no Yellow fever.

Response – Thank you, we have deleted Culex mosquitoes as advised

---

## [Decision Letter · Decision Letter 1]

17 Mar 2022

PONE-D-21-27941R1Malaria transmission in Africa: its relationship with yellow fever and measlesPLOS ONE

Dear Dr.  Oyeyemi,

Thank you for submitting your manuscript to PLoS ONE. After careful consideration, we feel that your manuscript will likely be suitable for publication if the authors revise it to address specific points raised now by the reviewer #1. According to the reviewer, there are some specific areas where further improvements would be of substantial benefit to the readers.   For your guidance, a copy of the reviewers' comments was included below. 

We look forward to receiving your revised manuscript.

Kind regards,

Luzia Helena Carvalho, Ph.D.

Academic Editor

PLOS ONE

Journal Requirements:

Reviewers' comments:

Reviewer's Responses to Questions

**Comments to the Author**

1. If the authors have adequately addressed your comments raised in a previous round of review and you feel that this manuscript is now acceptable for publication, you may indicate that here to bypass the “Comments to the Author” section, enter your conflict of interest statement in the “Confidential to Editor” section, and submit your "Accept" recommendation.

Reviewer #2: (No Response)

Reviewer #3: All comments have been addressed

2. Is the manuscript technically sound, and do the data support the conclusions?

Reviewer #2: Yes

Reviewer #3: Yes

3. Has the statistical analysis been performed appropriately and rigorously? 

Reviewer #2: Yes

Reviewer #3: Yes

4. Have the authors made all data underlying the findings in their manuscript fully available?

Reviewer #2: Yes

Reviewer #3: Yes

5. Is the manuscript presented in an intelligible fashion and written in standard English?

Reviewer #2: Yes

Reviewer #3: Yes

6. Review Comments to the Author

Reviewer #2: First, I want to congratulate the authors for implementing this relevance and timely research work. The study is important, well-written and it can provide insights to show the association of malaria transmission with common viral infection in Africa. However, while I goes across throughout the text, I found few concerns that that should be considered and needs to be modified prior to publication. So, I recommend the author to check my comments and suggestions which I indicated in the editable version of the PDF attached herein below.

As a general comment, I need the author to consider the following comments

1. The background the section of the manuscript is too shallow and the author incorporate few outdated figures as I indicated with the attached pdf file. So, I need the author to show the gab and the relevance of the work in detail.

2. Despite the method section is adequately expressed, still I need the author to consider my suggestions in the statistical analysis sub-headings of the text.

3. There are still few things that needs to be modified based on my suggestion as it indicated in the attached file.

4. Like that of the background section, the discussion of the manuscript is too shallow and the findings of the author is not discussed well. Moreover, the authors lack inconsistency during expressing the findings of the work. for example in the result section of the manuscript (line#197-198) the author indicated that the incidence of malaria is negatively associated with measles case, but this finding is reciprocally indicated in the discussion section of the manuscript (line #246-247). So, the the author should be discuss in detail for the each findings of the result of this work prior to submit the revised version of this text.

5. The author failed to show the strength and limitations of the study.

Reviewer #3: All comments, suggestions and questions made by the previous reviewer were addressed by the authors.

7. PLOS authors have the option to publish the peer review history of their article (what does this mean?). If published, this will include your full peer review and any attached files.

Reviewer #2: **Yes: **Tegegne Eshetu

Reviewer #3: No

---

## [Author Response · Author response to Decision Letter 1]

25 Mar 2022

Response to reviewer’s comments

Abstract

Comment - Do you think that having only few studies indicate the relevance of your study? Better to show existing gaps that assured the valuable of your study in this section?

Response – Thank you, we have modified the statement (see lines 32-33).

Comment - please Indicate data management and clearance methods if the author implemented prior to data analysis

Response – Thank you for the comment. While we did not include this because of words limit in abstract section, we instead given the detail in the materials and methods section

Comment - Please indicate the significant limit of your analysis.

Response – Thank you. This is now included (Line 39)

Comment - The author should conclude the main findings of this work before indicating the relevance of this work.

Response – Thank you, this is now revised accordingly

Introduction

The background section of your text is somewhat shallow. So, I suggest the author to re-write the background section of the text in detail based on the following structure if the author agreed with this structure. 

1. malaria is xxx and Malaria causes X and Y cases and problems globally and /or in africa..... Cuurently, WHO and/or other stakeholders recommend XXX public health intervention measures to control and prevent or eliminate or eradicate....malaria. 

2. However, the co-existence of XXX viral infection in similar geographical location with malaria 

3. Then if their any available challenges due to having similar geographical distribution..may in relation to current prevention and control mechanism, may be in relation to vaccine effectiveness....may be in related to treatment failure or anyother. 

Then gabs and the relevance your finding to fill such available gabs 

Response – Thank you for these important observations. We have included more information as suggested to include the gaps and the justifications for the study (see the red highlights).

Comments - I am feeling there is something cracked between the previous paragraph and this one. So, I recommend the author to incorporate few transition sentences prior to define yellow fever (i;e. a sentences which act as bridge between malaria you stated in the first paragraph and yellow fever in this paragraph to keep idea flow better. 

Response – Thank you, we have now rephrased for flow of idea (see Line 61)

Comment - Please add a reference for the impact of placental malaria on the infant immunity to measle.

Response – Thank you, reference now included (see ref. 15 in line 78).

Comment – ‘’Because all the studied variables are cases and rates which are regarded as count data, a Panel Poisson regression model was adopted to understudy these relationships. Where there are observed excess zero in the cases of event considered in the study, a zero-inflated Poisson (ZIP) regression that is robust to the occurrence of zero is adopted to achieve our specified objectives’’. Such statements should be incorporated in the statistical analysis sub-heading of the tesx

Response – We have now included the statement in the statistical analysis sub-section.

Materials and methods (line 167-171).

Comments - How could manage and associate the findings of malaria with yellow fever and measles in this paper since the author collected the recorded data in different time period. So, did the author consider this time period variation while the author conclude the association of malaria incident and death with yellow fever and measles? 

Response: Thank you for your comments

There are two main analyses in the study. The first analysis was the determination of annual trend change for each Malaria incidence, Death due to malaria, Measles and Yellow fever. Here, changes of these four variables were evaluated over time for each country considered in the study and each dataset were treated separately for this analysis. The time horizon for malaria incidence and death due to the malaria trend equation was 2000-2018 whereas the time horizon for Measles and Yellow fever trend equation was 1980 and 2019. This decision was based on the availability of the data obtained from the secondary source mentioned in the manuscript.

The second analysis was a regression equation where the effect of both Measles and Yellow fever on each of the Malaria incidence and Death due to malaria was investigated. Two data files were used for this analysis bearing in mind that in any regression problem there must be a pair of x and y variables. The first data file had Malaria as the dependent variable while both Measles and Yellow fever were the independent variables. The time horizon for this regression was 2000-2018 since information for malaria incidence was not available for 1980-1999 and 2019. Similarly, the second data file contained Death due to malaria as the dependent variable with Measles and Yellow fever as independent variables. The time horizon for this analysis was 2000-2017

Comment - Despite you are indicating the significance and insignificance of your finding in your result and discussion sub-heading, the author failed to indicate the significance limit of your analysis. So please add here the significance limit for your statistical analysis output. 

Response – Thank you for the observation. We have now included the significance limit (see line 173).

Results

Comment – Figures quality should be improved

Response - Thank you. We have now used the PACE website provided by Plos to help ensure that figures meet PLOS requirements and improve the quality.

Discussion

Comment - According to the recent World Health Organization (WHO) published report showed, malaria and related deaths have been increased after 2019. If so, How could you harmonize your conclusion based on your finding with the WHO report?

Response – Thank you for this important observation. We have tried to provide reasons for your concern in the MS as follow “It is however important to stress that the recent WHO published report showed increase in malaria cases and associated deaths after 2019 [27]. The reasons for this have been accrued to plateaued malaria interventional funding, increase in drug and insecticide resistance, and climate change which threatens to push malaria transmission to new regions [28]. COVID-19 has further disrupted progress and implementation of various malaria intervention programs [28]”. Please see lines 256-260.

Comment - Are sure does modification of mosquito biting behavior is a factor for decline of malaria? I need more justification for this statement if your answer is yes.

Response – Thank you for your comment. This is an observation from another author which we referenced. Yes, we agree with the statement as the inability of the mosquitoes to access humans for blood meal indoor due to hindrance caused by ITN could trigger a behavioural change in mosquitoes to source for blood meal outside where there is no ITN.

Comment - would you mention few socio-ecological conditions which favour for the reduction of density of mosquito? Please elaborate it by giving example?

Response – Thank you for the observation. The examples are already given in the text. Please see the sentence below.

“changes in socio-ecological conditions in many malaria-endemic areas of Africa including changes in temperature, deforestation [22], increase use of agricultural pesticides with mosquitocidal properties [23], improved house constructions [24] could result in a decreased density of mosquitoes”.

Comment - Your justification is insuffeicent? please discuss in detail the possible reasons why malaria associated deaths is high in Africa unlike other past of the coninent.

Response – Thank you for the comment. However, the factors already highlighted are the major reasons we believe to be associated with high malaria deaths in Africa.

Comment - Are you sure do the incidence of malaria and measles case showed a positive relationship based on your finding? But you indicated in line 197-198, o]a significant negative relationship existed between malaria and measles cases (P<0.05). If so, why the author failed to keep the consistence of the findings? The author should give a clear information for this contradictory conclussion? Moreover, the author expected to discuss possible reasons why this findings happen?

Response – Thank you for this very important observation. This is a mistake from our end. We have now corrected it. We have also changed the direction of the discussion based on this change (see lines 276-287).

Comment - Why the author need to discuss the incidence of malaria in association with HIV and hebatitis? Does this is the authors objective? How could you relate it?

Response – Thank you for the observation. Since there were no enough reports to discuss the association between malaria and the studied vaccine related viruses we thought discussing the relationships between malaria and other viral agents could shed some light on the relationships between malaria and measles/YFV. Since the reviewer is not satisfied with this, we have now expunged the sentences and the references.

Comment - Are you sure does the region shared 30% of malaria associated death?

Response – Thank you for the comment. Based on the reference WHO report cited, the total malaria-related deaths in West Africa Region is more than 30%. Nigeria alone accounts for 25%.

---

## [Decision Letter · Decision Letter 2]

31 Mar 2022

PONE-D-21-27941R2Malaria transmission in Africa: its relationship with yellow fever and measlesPLOS ONE

Dear Dr. Oyeyemi,

Thank you for submitting your manuscript for review to PLoS ONE. After careful consideration, we feel that your manuscript will likely be suitable for publication if the authors revise it to address a critical point previously raised by the reviewer.  According to reviewer #2, the authors must specify the exact statistical output of P-value rather than indicated as < 0.05 in the table 1. This information is critical to the reliability of the data analysis.

We look forward to receiving your revised manuscript.

Kind regards,

Luzia Helena Carvalho, Ph.D.

Academic Editor

PLOS ONE

Journal Requirements:

Reviewers' comments:

Reviewer's Responses to Questions

**Comments to the Author**

1. If the authors have adequately addressed your comments raised in a previous round of review and you feel that this manuscript is now acceptable for publication, you may indicate that here to bypass the “Comments to the Author” section, enter your conflict of interest statement in the “Confidential to Editor” section, and submit your "Accept" recommendation.

Reviewer #2: All comments have been addressed

2. Is the manuscript technically sound, and do the data support the conclusions?

Reviewer #2: Yes

3. Has the statistical analysis been performed appropriately and rigorously? 

Reviewer #2: Yes

4. Have the authors made all data underlying the findings in their manuscript fully available?

Reviewer #2: Yes

5. Is the manuscript presented in an intelligible fashion and written in standard English?

Reviewer #2: Yes

6. Review Comments to the Author

Reviewer #2: All most all points I raised as a reviewer were satisfactorily addressed by the authors. I congratulate the authors on this achievement. However, there is only one major issue that I need to bring up again.

1. Why did the authors failed to specify the exact statistical output of P-value rather than indicated as < 0.05 in the table 1? Unless and until the author provides a satisfactory explanation for why the P-value was not indicated, I will be skeptical of the data analysis's reliability.

7. PLOS authors have the option to publish the peer review history of their article (what does this mean?). If published, this will include your full peer review and any attached files.

Reviewer #2: **Yes: **Tegegne Eshetu

---

## [Author Response · Author response to Decision Letter 2]

1 Apr 2022

Why did the authors failed to specify the exact statistical output of P-value rather than indicated as < 0.05 in the table 1? Unless and until the author provides a satisfactory explanation for why the P-value was not indicated, I will be skeptical of the data analysis's reliability.

Response – Thank you for your comment. We merely supplied the level of significance (P<0.05 or P>0.05) in the previous version. However, we have now provided the exact P value as requested by the reviewer.

---

## [Decision Letter · Decision Letter 3]

18 Apr 2022

PONE-D-21-27941R3Malaria transmission in Africa: its relationship with yellow fever and measlesPLOS ONE

Dear Dr. Oyeyemi, Thank you for submitting your manuscript for review to PLoS ONE. After careful consideration, we feel that your manuscript will likely be suitable for publication if the authors revise it to address critical points raised by the reviewer. According to reviewer, there are some specific areas where further improvements would be of substantial benefit to the readers. A copy of the reviewers’ comments was included for your information.

Please submit your revised manuscript by April 30 If you will need more time than this to complete your revisions, please reply to this message or contact the journal office at plosone@plos.org. Please include the following items when submitting your revised manuscript:A rebuttal letter that responds to each point raised by the academic editor and reviewer(s). You should upload this letter as a separate file labeled 'Response to Reviewers'.A marked-up copy of your manuscript that highlights changes made to the original version. You should upload this as a separate file labeled 'Revised Manuscript with Track Changes'.An unmarked version of your revised paper without tracked changes. You should upload this as a separate file labeled 'Manuscript'.If applicable, we recommend that you deposit your laboratory protocols in protocols.io to enhance the reproducibility of your results. Protocols.io assigns your protocol its own identifier (DOI) so that it can be cited independently in the future. For instructions see: https://journals.plos.org/plosone/s/submission-guidelines#loc-laboratory-protocols. Additionally, PLOS ONE offers an option for publishing peer-reviewed Lab Protocol articles, which describe protocols hosted on protocols.io. Read more information on sharing protocols at https://plos.org/protocols?utm_medium=editorial-email&utm_source=authorletters&utm_campaign=protocols.

We look forward to receiving your revised manuscript.

Kind regards,

Luzia Helena Carvalho, Ph.D.

Academic Editor

PLOS ONE

Reviewers' comments:

Reviewer's Responses to Questions

**Comments to the Author**

1. If the authors have adequately addressed your comments raised in a previous round of review and you feel that this manuscript is now acceptable for publication, you may indicate that here to bypass the “Comments to the Author” section, enter your conflict of interest statement in the “Confidential to Editor” section, and submit your "Accept" recommendation.

Reviewer #2: All comments have been addressed

2. Is the manuscript technically sound, and do the data support the conclusions?

Reviewer #2: Yes

3. Has the statistical analysis been performed appropriately and rigorously? 

Reviewer #2: Yes

4. Have the authors made all data underlying the findings in their manuscript fully available?

Reviewer #2: Yes

5. Is the manuscript presented in an intelligible fashion and written in standard English?

Reviewer #2: Yes

6. Review Comments to the Author

Reviewer #2: First and foremost, I'd like to thank the authors for their success in producing a scientifically soundable article while taking into account my previous comments and suggestions. Having said that, I'd like to make a suggestion for the authors in table one.

In table one, when the author made a change after considering my previous comment, the p-value was written as o.ooo, which is not a scientifically sound way of writing. To the best of my knowledge, P-value cannot be written as 0.000 as you did in table 1. Even if the statistical software produces a p value of 0.000, the exact value cannot be equals to zero. You can verify it by double-clicking on it, which will display the actual value of P. In such cases, P value should be expressed as < o.oo1. The logic behind for such expression is the observed value can never be zero due to the presence of random error, which is unavoidable and uncontrollable. Thus, I recommend the authors to express the P-value as < 0.001 in table 1 rather than simply put 0.000. However, it should only be done if and only if the author agrees with my comment.

7. PLOS authors have the option to publish the peer review history of their article (what does this mean?). If published, this will include your full peer review and any attached files.

Reviewer #2: No

---

## [Author Response · Author response to Decision Letter 3]

18 Apr 2022

Response to reviewer

Reviewer #2: First and foremost, I'd like to thank the authors for their success in producing a scientifically soundable article while taking into account my previous comments and suggestions. Having said that, I'd like to make a suggestion for the authors in table one.

In table one, when the author made a change after considering my previous comment, the p-value was written as o.ooo, which is not a scientifically sound way of writing. To the best of my knowledge, P-value cannot be written as 0.000 as you did in table 1. Even if the statistical software produces a p value of 0.000, the exact value cannot be equals to zero. You can verify it by double-clicking on it, which will display the actual value of P. In such cases, P value should be expressed as < o.oo1. The logic behind for such expression is the observed value can never be zero due to the presence of random error, which is unavoidable and uncontrollable. Thus, I recommend the authors to express the P-value as < 0.001 in table 1 rather than simply put 0.000. However, it should only be done if and only if the author agrees with my comment.

Response – Thank you for this important comment. We agree with the reviewer on this point. So, we have now replaced P value 0.000 with <0.001 as suggested.

---

## [Editor Report · Decision Letter 4]

22 Apr 2022

Malaria transmission in Africa: its relationship with yellow fever and measles

PONE-D-21-27941R4

Dear Dr. Oyeyemi,

We’re pleased to inform you that your manuscript has been judged scientifically suitable for publication and will be formally accepted for publication once it meets all outstanding technical requirements.

Kind regards,

Luzia Helena Carvalho, Ph.D.

Academic Editor

PLOS ONE